# Hybrid Analytical Platform Based on Field-Asymmetric Ion Mobility Spectrometry, Infrared Sensing, and Luminescence-Based Oxygen Sensing for Exhaled Breath Analysis

**DOI:** 10.3390/s19122653

**Published:** 2019-06-12

**Authors:** L. Tamina Hagemann, Stefan Repp, Boris Mizaikoff

**Affiliations:** Institute of Analytical and Bioanalytical Chemistry (IABC), Ulm University, Albert-Einstein-Allee 11, 89081 Ulm, Germany; tamina.hagemann@uni-ulm.de (L.T.H.); stefan.repp@uni-ulm.de (S.R.)

**Keywords:** exhaled breath analysis, field-asymmetric ion mobility spectrometry (FAIMS), Fourier-transform infrared spectroscopy (FTIR), luminescence sensing, infrared sensors, hyphenated techniques, hybrid techniques, acetone, carbon dioxide, oxygen

## Abstract

The reliable online analysis of volatile compounds in exhaled breath remains a challenge, as a plethora of molecules occur in different concentration ranges (i.e., ppt to %) and need to be detected against an extremely complex background matrix. Although this complexity is commonly addressed by hyphenating a specific analytical technique with appropriate preconcentration and/or preseparation strategies prior to detection, we herein propose the combination of three different detector types based on truly orthogonal measurement principles as an alternative solution: Field-asymmetric ion mobility spectrometry (FAIMS), Fourier-transform infrared (FTIR) spectroscopy-based sensors utilizing substrate-integrated hollow waveguides (iHWG), and luminescence sensing (LS). By carefully aligning the experimental needs and measurement protocols of all three methods, they were successfully integrated into a single compact analytical platform suitable for online measurements. The analytical performance of this prototype system was tested via artificial breath samples containing nitrogen (N_2_), oxygen (O_2_), carbon dioxide (CO_2_), and acetone as a model volatile organic compound (VOC) commonly present in breath. All three target analytes could be detected within their respectively breath-relevant concentration range, i.e., CO_2_ and O_2_ at 3-5 % and at ~19.6 %, respectively, while acetone could be detected with LOQs as low as 165-405 ppt. Orthogonality of the three methods operating in concert was clearly proven, which is essential to cover a possibly wide range of detectable analytes. Finally, the remaining challenges toward the implementation of the developed hybrid FAIMS-FTIR-LS system for exhaled breath analysis for metabolic studies in small animal intensive care units are discussed.

## 1. Introduction

Exhaled breath contains a wide variety of molecules that are potentially useful for noninvasive therapy monitoring and elucidation of metabolic pathways [1,2,3,4,5,6,7]. However, breath analysis comes with a variety of challenges that need to be met by corresponding analytical tools. The complexity of exhaled breath samples is based on numerous molecular components [8] that vary widely in their chemical nature (i.e., organic/inorganic, polarity, ionizability, etc.) and in their concentration (i.e., from ppt for selected volatile compounds to % for CO_2_ and O_2_) [3,8]. The resulting sample complexity is further aggravated by high humidity levels [6] present in breath affecting some of the commonly applied analytical techniques such as GC-MS. Furthermore, interpatient variance based on differences in, e.g., age, diet, medication, smoking habits, or use of sanitary articles [6], as well as the influence of the composition of the inhaled air [9], are difficult to address. Moreover, despite its increasing importance, the field of breath analysis still lacks in commonly accepted guidelines for standardized sampling, analysis procedures, and data handling. Hence, comparability and reproducibility of the obtained results is frequently limited [3,4,10].

Due to the inherent complexity, it is almost impossible to comprehensively analyze breath samples with single analytical techniques. Therefore, hyphenated approaches combining a molecularly selective device (e.g., mass spectrometry (MS) [11,12,13] or ion mobility spectrometry (IMS) [14,15,16]) with preconcentration schemes (e.g., via solid-phase microextraction (SPME) or needle trap devices (NTD) [4,17,18]) and/or pre-separation strategies (e.g., via gas chromatography (GC) or multi-capillary columns (MCC) [3,14,15,16,19]) are considered the state-of-the-art for addressing trace concentrations and to reduce sample complexity. Yet, in most cases, only one type of ‘detector’ (e.g., MS or IMS) is still used, limiting the number of detectable breath analytes. For example, most IMS systems lack the ability to detect alkanes [20], which are an important class of breath analytes (e.g., ethane, pentane, etc.) [5,21,22]. In turn, MS systems may, in fact, detect a broad range of analytes, but are expensive, bulky, and usually not mobile [7], which renders their applicability as compact and portable breath analyzer directly at the point-of-care (POC) limited.

An alternative approach towards covering a possibly broad range of exhaled breath constituents is to hyphenate several detector types, which was the goal of the present study. Only a few research groups have proposed [23,24] or selected (see Table 1) this path. Orthogonal detection techniques have been successfully combined [25,26,27,28,29,30], however, not into a single device.. For example, Covington et al. [25] applied an electronic nose (enose) and GC-IMS for analyzing the same breath sample, and Williams et al. [29] used nondispersive infrared (NDIR) analysis and proton-transfer-reaction time-of-flight mass spectrometry (PTR-TOF-MS) in parallel for the same set of samples. However, the applied analytical devices were still used as standalone techniques. This entails extensive sample handling with potentially associated handling errors and extended analysis times, e.g., required for separate sample injection individually into each device. As a result, the application in POC scenarios again appears limited. 

Monks et al. [31] published an approach entailing simultaneous flame ionization detection (FID) and quadrupole MS analysis via a truly integrated GCxGC-FID/qMS setup. However, only offline breath analysis was facilitated, which is also a disadvantage of nearly all aforementioned combinations. Offline breath analysis frequently involves gas bags or sample storage, thus taking the risk of cross-contamination and/or sample degradation [3], which is of particular importance if trace breath analytes have to be detected. 

A common approach enabling online analysis is the use of electronic noses [32,33,34,35,36,37,38,39], i.e., arrays of a variety of sensing principles, such as chemresistors or quartz microbalances coated with–in part conductive–polymer membranes individually responding to different types of molecules. Although such sensor arrays offer portable and rapidly responding breath detection capabilities, specific biomarker identification and inter-device comparability remain problematic [40]. Vaks et al. [41] and Shorter et al. [42] selected a different approach by combining light sources emitting complementary wavelengths or wavelength regimes (i.e., subTHz, THz, IR, etc.) into a single setup, demonstrating, at least partially, online analysis [42]. However, while the usage of different wavelength regimes provides some orthogonality, the basic detection mechanism was essentially the same, i.e., molecules not responding to optical interrogation (sufficient light absorption in the selected wavelength range) will remain undetected. This underlines the necessity to choose truly orthogonal methods, which are based on different physical stimuli generating the analytical signals. 

Miekisch et al. [43,44] presented a multidimensional sensing platform including hemodynamic monitoring as well as breath monitoring via capnometry, spirometry and PTR-TOF-MS, i.e., truly orthogonal techniques integrated into a single online monitoring platform. However, while this is an impressively comprehensive approach for breath analysis, the addressed breath volumes are suitable for the analysis of human breath, yet limited for monitoring the minute breath sample volumes (i.e., hundreds pf microliters) in small animal models such as mice, which is the focus of the present study.

The long-term goal founded on the present study is to develop a comprehensive sensing platform based on truly orthogonal detection principles for online breath analysis in a mouse intensive care unit (MICU) operated at the Institute of Anesthesiological Pathophysiology and Method Development (IAPMD) at Ulm University Medical Center to gain insight into the metabolism of mice with induced trauma aiding the development of therapeutic approaches. 

Given its size, instrumental space available in a MICU is limited rendering the implementation of, e.g., PTR-TOF-MS, as used by Miekisch et al. [43,44] (width x height x length: 56x130x78 cm) difficult. Furthermore, the setup of Miekisch et al. was connected to the patient’s mouthpiece via a 6 m-long tube, avoiding safety risks potentially arising from interferences between the analytical setup and the medical equipment [45]. This further increases the device footprint and causes a significantly large dead volume, which is problematic because implementation at a MICU—and specifically at Ulm University Medical Center—is limited by the specific ventilation scheme of the mouse: In order to prevent the mouse alveoli from collapsing, a positive end-expiratory pressure (PEEP) valve is a major component of the mechanical ventilation system. In order for the PEEP valve to work properly, the dead volume of the entire system needs to be kept at a minimum.

The analysis of ^12^CO_2_, ^13^CO_2_, and O_2_ concentrations, as well as the respiratory quotient (RQ) in mouse breath, has already been enabled by our research team via various analytical tools (i.e., iHWG-FTIR spectroscopy, interband cascade laser based tunable diode laser absorption spectroscopy (TDLAS), and LS) [46,47,48]. Besides these already quantifiable analytes in mouse breath, the detection of additional volatile compounds such as acetone and H_2_S is desirable for therapy monitoring and to aid in understanding the underlying metabolism of traumatized mice. 

The approach proposed herein is therefore based on integrating a field-asymmetric ion mobility spectrometer (FAIMS) for volatile compound detection with our previously developed iHWG-FTIR spectroscopy and LS setup into a compact breath analysis system (see Figure 1). These three methods were selected, as they promise excellent detection orthogonality, i.e., the underlying physical concepts are complementary: While FAIMS separates and detects ionized analytes based on the difference in ion mobility at high and low field conditions, iHWG-FTIR and LS address molecular vibrations and rotations, as well as ro-vibrations in the infrared wavelength regime, and luminescence quenching of an organic dye in the presence of oxygen, respectively. The detection principles were described in more detail elsewhere (i.e., FAIMS [49], iHWG-FTIR spectroscopy [50], LS [51]). 

In order to implement a combined FAIMS-FTIR-LS system in a MICU, several obstacles have to be overcome. The selected methods have different experimental requirements, e.g., in terms of gas flow and pressure, measurement protocols, background recordings, and duration of individual measurements. These requirements need to be aligned for enabling simultaneous measurements in a combined system in an online monitoring scenario. In addition, the challenges common to breath analysis in general including, e.g., the detection of trace concentrations despite high humidity levels and complex background matrices need to be addressed. Last but not least, as already mentioned, additional challenges arise from the fact that sensing mouse breath requires sampling and analyzing small exhaled breath volumes (i.e., few hundreds of microliters) such that the mouse ventilation scheme is not negatively affected. 

As a first step, the present study aims at demonstrating the fundamental feasibility and complementarity of the selected methods, along with their experimental integration into a combined FAIMS-FTIR-LS system despite the different sampling requirements. In addition, it was anticipated demonstrating that the sensitivity of each technique is sufficient for targeting breath-relevant concentrations of the individual analytes in a truly orthogonal fashion, i.e., without interferences between the individual techniques. Hence, after implementation of FAIMS, FTIR and LS into a single system, synthetic breath samples containing acetone (as an exemplary yet relevant volatile compound in breath [52]), CO_2_, O_2_, and N_2_ were prepared and analyzed to demonstrate full functionality of the developed hyphenated prototype. The resulting data clearly corroborated the feasibility and utility of the integrated analytical platform for simultaneous online analysis of synthetic breath samples. It was also shown that the detection of all analytes in the respective breath-relevant concentration range was enabled, and that FAIMS, FTIR, and LS work orthogonally and without interference.

## 2. Materials and Methods

### 2.1. Hybrid Analytical Platform

#### 2.1.1. Gas Sample Preparation

A stock gas mixture of 2.33 ppm acetone in synthetic air (± 0.23 ppm, MTI Industriegase, Neu-Ulm, Germany) was diluted down by synthetic air (produced with 20.5 vol.% O_2_ grade 5.0, remains N_2_ grade 5.0, H_2_O ≤ 5 ppmv, NO+NO_2_ ≤ 0.1 ppmv, low molecular weight hydrocarbons C*_n_*H*_m_* < 0.1 ppmv, by MTI Industriegase, Neu-Ulm, Germany) and CO_2_ (technical grade (DIN EN ISO 14175), ≥ 99.8 vol-%, N_2_ ≤ 1000 ppmv, H_2_O ≤ 120 ppmv, MTI Industriegase, Neu-Ulm, Germany) to eight samples, containing acetone concentrations between 0 and 23 ppb and a background concentration of 3%, 4%, or 5% CO_2_ and 19.6% ± 0.5% O_2_ (concentrations given here are volumetric concentrations). The acetone, air, and CO_2_ flows were regulated by mass flow controllers (Bronkhorst El Flow Prestige, FG-201CV-RBD-11-K-DA-000, 80 mL/min full scale capacity for acetone; FG-201CV-ABD-11-V-DA-000, 3000 mL/min full scale capacity for synthetic air; Vögtlin red-y smart series, type GSC-A9KS-BB22, 200 mL/min full scale capacity for CO_2_). For cleaning and drying purposes, air and CO_2_ were filtered through active charcoal (# 20626, Restek, Bad Homburg, Germany), molecular sieve (5Å pore size, # 8475.2, Carl Roth GmbH & Co KG, Karlsruhe, Germany), and sintered glass filter elements (VitraPor^®^, 40–100 µm, 4–5.5 µm, 1.5 µm). The dewpoint of air and CO_2_ was measured to be −39.8 °C (humidity sensor SF52-2-X-T1-B, Michell Instruments, Ely, UK), corresponding to a water content of 192 ppm. The acetone sample gas was neither VOC filtered nor dried, since this would have caused analyte loss. The water content in the acetone gas cylinder was assumed to be negligible due to the dilution of acetone sample gas in comparatively big volumes of CO_2_/air. 

Acetone and CO_2_ were mixed first, by leading their flow through a filter with 0.5 μm pore size (SS-2TF-05, Swagelok, Reutlingen, Germany) to induce turbulences for homogeneous mixing. The combined acetone/CO_2_ flow was then combined with the air flow. A schematic of the gas mixing unit is displayed in Figure 2 (left half) in Section 2.1.2 together with the hybrid FAIMS-FTIR-LS sensing platform. 

#### 2.1.2. Hybrid FAIMS-FTIR-LS Platform and Concentration-Dependent Measurements

The hybrid analytical platform is displayed in Figure 2 and experimental details are provided in Table 2. Gas samples were provided by the gas mixing unit displayed in the left half of Figure 2 and described in the previous section. The sample flow produced by the gas mixing unit was constantly kept at 2200 mL/min. The relief valve (SS-RL3S4, Swagelok, Reutlingen, Germany) between the gas mixing unit and the FTIR/O_2_ sensor unit was adjusted so that the flow reaching the FTIR/O_2_ sensor unit was 400 ± 10 mL/min and the flow through the FAIMS PAD was 1800 ± 30 mL/min. These flows were regularly checked on with a digital flow meter (ADM1000, J&W Scientific, Folsom, CA, USA) at the outlet of the O_2_ sensor and with the flow sensor integrated in the FAIMS PAD, respectively. To minimize analyte adsorption along the tubing walls, perfluoroalkoxy alkane (PFA) tubings (1/8’’ and 1/4’’ outer diameter, Swagelok, Reutlingen, Germany) and heated (41 °C) Sulfinert tubings (#29242, Restek, Bad Homburg, Germany) were used in order to minimize analyte adsorption.

Before starting a measurement series, a hold time was adopted until flow and pressure had stabilized in the FAIMS device (1800 ± 30 mL/min, 0.800 ± 0.020 bar*g*) to ensure reproducibility of the FAIMS data. In case the flow and pressure varied beyond the given limits, the needle valve at the exhaust of the FAIMS, as well as the relief valve between FAIMS and FTIR, were adjusted until flow and pressure had stabilized for at least ten minutes in the range defined above. 

Each measurement series included eight acetone/CO_2_/air gas samples. Prior to the analysis of an acetone/CO_2_/air mixture, one sample containing pure air and one sample containing only air and CO_2_ were recorded (see Table 2). During the pure air sample, the FTIR background was recorded, and the according FAIMS spectrum was used to ensure that the system had entirely cleaned down after the previous sample (Cleanliness criterion: Maximum contamination peak intensity at 54% DF in the CV interval from −0.1V to +0.2V had to be ≤ 0.08 a.u. prior to each consecutive measurement). The CO_2_/air measurement, on the other hand, served as a background spectrum for FAIMS. Before analysis, the respective sample gas was led through the setup for at least two minutes to ensure a constant analyte concentration in the whole setup and during the entire measurement.

After recording a blank as the first sample of every measurement series, the remaining samples were analyzed in a random sample order that was different in each measurement series. The CO_2_ concentration was constant (3%, 4%, or 5%) within one measurement series. Three measurement series were recorded per CO_2_ concentration. For all air, CO_2_/air and all acetone/CO_2_/air samples, five FAIMS spectra (~19 min), and five FTIR spectra (~ 3.5 min) were successively recorded, while the sample was continuously flowing through the hybrid setup. Simultaneously, the O_2_ concentration was continuously monitored for the duration of the FAIMS measurements. 

### 2.2. Details on the Individual Analytical Methods

#### 2.2.1. Field-Asymmetric Ion Mobility Spectrometry

FAIMS data were recorded with an OEM FAIMS PAD (Owlstone Inc., Cambridge, UK), using the Lonestar software (version 4.912, Owlstone Inc., Cambridge, UK). After ionization by a ^63^Ni ionization source, analytes were detected by the FAIMS sensor (gap size 37 µm; RF waveform: 267 ±2 V maximum peak-to-peak voltage, 26 MHz ± 26 Hz RF, 25% Duty Cycle, 51 steps; compensation voltage (CV) from −6 to +6 V (512 steps, ~4.5 s per full CV scan), flow 1800 ± 30 mL/min; sensor temperature: 60 °C). The sample gas was continuously flowing through the spectrometer at 1800 ± 30 mL/min as the data were recorded. The pressure could be regulated via the needle valve (SS-2MG-MH, Swagelok, Reutlingen, Germany) at the FAIMS outlet and was set to 0.800 ± 0.020 bar*g*. A membrane filter at the inlet of the FAIMS device (polytetrafluoroethylene (PTFE) membrane, 1 µm pore size), heated to 100 °C to avoid analyte accumulation in the filter, prevented particle introduction into the FAIMS PAD. In order to avoid the build-up of charge, the intersweep delay between two subsequent recordings was set to 1500 ms. The obtained FAIMS spectra, also called dispersion plots, displayed the ion current on the detector (z axis) in dependence on the CV (x axis) and the percentage of the dispersion field (DF), which was scanned by varying the peak-to-peak-voltage between 0 and 267 V stepwise. 

#### 2.2.2. Substrate-Integrated Hollow Waveguide Coupled Fourier-Transform Infrared Spectroscopy

CO_2_ concentrations were recorded via iHWG coupled FTIR spectroscopy. The setup and gas cell are described in detail elsewhere [53]. Light from an ALPHA FTIR spectrometer (Bruker Optik GmbH, Ettlingen, Germany) was coupled into an iHWG (aluminum, 7.5 cm optical path length, 4 mm × 4 mm internal cross-section, produced by fine mechanical workshop West, Ulm University, Ulm, Germany), and then onto the internal detector of the spectrometer via two gold-coated off-axis parabolic mirrors (Thorlabs, MPD254254-90-M01, 2″ RFL). Using the software OPUS (version 7.2, Bruker Optik GmbH, Ettlingen, Germany), IR spectra were recorded in the spectral range from 4000 to 400 cm^−1^ at a spectral resolution of 2 cm^−1^, with 20 averaged scans, and at a flow rate of 400 ± 10 mL/min. The Fourier transformation was done in OPUS, using the Blackman-Harris three-term apodization function. In order to exclude CO_2_ from ambient air from the optical absorption paths, the entire IR setup was housed in a plastic bag which was purged with synthetic air for at least 15 minutes prior to, as well as during, each measurement series. 

#### 2.2.3. Oxygen Sensing

A flow-through O_2_ sensor, detecting O_2_ based on luminescence quenching, (FireStingO2, Pyro Science GmbH, Aachen, Germany) [51] was used for monitoring the O_2_ concentration, supported by the Software FireSting Logger (version 2.365, PyroScience GmbH, Aachen, Germany). One data point per second was recorded. 

### 2.3. Data Processing

#### 2.3.1. Field-Asymmetric Ion Mobility Spectrometry 

Since a direct import of the FAIMS data (.dfm format) into Matlab was not possible, FAIMS data were exported from the Lonestar software as text files and then imported into Matlab (R2018A, The Mathworks Inc., Natick, MA, USA). For baseline correction, the average of the five repetitions of the FAIMS dispersion plot of an CO_2_/air sample was subtracted from the average of the five repetitions of the subsequent acetone CO_2_/air sample. Acetone monomer and dimer peak volumes were approximated by respectively summing all intensity values in selected regions of the dispersion plot (monomer: 68% to 72% DF, −2.75 to −1.95 V CV; dimer: 46% to 50% DF, −0.35 to +0.45 V CV). These integration windows were chosen equally wide for monomer and dimer peak and based on a compromise between achievable signal height and freedom from interferences with other spectral components. The so-obtained monomer and dimer peak volumes were then added together to obtain the total acetone signal (from now on, only called “acetone signal”). Singly integrating the monomer or the dimer peak would have distorted the FAIMS data evaluation: While the monomer peak was very faint or even invisible at higher acetone concentrations, its contribution to the total acetone signal at higher concentrations would not have been negligible. 

After normalization with the mean acetone signal at the maximum measured acetone concentration (23 ppb), the signal was averaged and the standard deviation was calculated. The normalized and averaged acetone signal was plotted against the acetone concentration and an asymptotic fit (*y* = *A* − *B·C^x^*) was applied. Following IUPAC regulations [54], the concentration at the limit of detection (LOD) and at the limit of quantification (LOQ) was estimated by inserting the signal intensity at the LOD and the LOQ (µ_B_ + 3.29·σ_B_ and µ_B_ + 10·σ_B_, respectively, with average normalized signal intensity of the blank µ_B_ and its according standard deviation σ_B_) into the inverse of the calibration function (*x* = ln((*A* − *Y*)/*B*)/ln*C*). 

#### 2.3.2. Fourier-Transform Infrared Spectroscopy 

IR data were imported from OPUS into Origin Pro 2017G. An exemplary spectrum is shown in Figure A1 in the Appendix A. For baseline correction, each IR spectrum was shifted by the median of the data set, since the latter suitably represented the baseline. The area under the baseline-corrected IR peak at 2360 cm^−1^ between 2200 and 2450 cm^−1^ was averaged for the five repetitions recorded in a row for each sample. The so obtained CO_2_ signal was then normalized by division by the overall maximum CO_2_ signal and the normalized signal was averaged for the three repetitions of the measurement series recorded for each CO_2_ concentration (3%, 4%, and 5% CO_2_). The according standard deviation was calculated. 

#### 2.3.3. Oxygen Sensing

For each measurement, the O_2_ concentrations directly output by the FireSting Logger software was averaged for the time span between 5 and 15 min after starting the O_2_ measurement. O_2_ concentrations recorded between 0 and 5 min were not included in the average because the O_2_ concentration reached an equilibrium after approximately 5 min (see Figure A2 in the Appendix A). The so obtained O_2_ signal was then normalized by division by the overall mean O_2_ signal. The normalized signal was averaged for the three repetitions of the measurement series recorded for each CO_2_ concentration and the standard deviation was calculated. 

## 3. Results and Discussion

### 3.1. FAIMS Results

As mentioned above, FAIMS dispersion plots of pure air and of CO_2_/air were recorded before recording an acetone/CO_2_/air containing sample (for further detail, see Table 2 in Section 2.1.2). Figure 3 exemplarily shows a dispersion plot for each sample type collected in positive mode. 

The dispersion plot of pure air (Figure 3a) mainly showed the reactant ion peak (RIP), which, in positive detection mode, appears due to the formation of ionized clusters of water molecules present in the carrier gas [55]. The faint vertical signal in Figure 3a at around 0 V CV was approximately constant for all recorded dispersion plots. Due to the immense sensitivity of FAIMS, this trace could not be erased throughout the whole project and was likely to be caused by substances emitted from the tubing and the FAIMS device itself. Following the instructions of this particular FAIMS system, this signal was used to establish a cleanliness criterion, i.e., the system was considered to be ready for reliable analysis only if the maximum peak intensity between −0.1 to +0.2 V CV at 54% DF was ≤ 0.08 a.u. 

The CO_2_/air dispersion plot (Figure 3b) also mainly showed the RIP. No clear analyte peak appeared, as expected, since CO_2_ is not ionizable by the ^63^Ni source. The faint additional trace at ~ 65% DF and −3 V CV presumably occurred because of contaminations from the CO_2_ gas bottle that could not be entirely removed by the used filters. The acetone/CO_2_/air dispersion plot (Figure 3c) showed an intensity decrease of the RIP, as well as the appearance of two main additional peaks. Generally, once an ionizable analyte like acetone is inserted into the FAIMS, one or two water molecules in the ionized carrier gas clusters are replaced by the analyte molecules. Hence, the RIP intensity decreases and a monomer and/or dimer peak appear, respectively. The tentative assignment of monomer and dimer peak, as it is indicated in Figure 3c, was based on the concentration-dependent behavior of both peaks: While the monomer peak intensity showed an intensity maximum at lower concentrations, the dimer peak constantly increased with increasing acetone concentration, as an additional water molecule in each monomer cluster was replaced by a second acetone molecule, thus forming a dimer cluster. The relative position of monomer and dimer peak further substantiated the peak assignment: the lighter, less bulky and hence more mobile monomer cluster gave rise to a peak at a lower CV than the less mobile dimer cluster. The exact origin of the faint feature between monomer and dimer peak in Figure 3c,d (~50%DF, -0.5 V CV) is unknown, but its potential effect is commented on later in this section. In order to obtain the net monomer and dimer signal, Figure 3b was subtracted from Figure 3c for background subtraction. The resulting data is shown in Figure 3d. The *z* axis of Figure 3d was varied compared to Figure 3a–c in order to make the monomer and dimer peak more clearly visible. At the position where the RIP appeared in Figure 3a–c, the signal intensity was negative in Figure 3d, since the RIP intensity decreased while acetone was present in the FAIMS sensing region. 

After averaging and normalization, the sum of monomer and dimer peak volume was plotted against the acetone concentration for 3%, 4%, and 5% CO_2_, respectively (see Figure 4a). Due to saturation of the FAIMS detector, the acetone signal converged toward a maximum value for higher acetone concentrations. Thus, an asymptotic fit was applied and the concentrations at the limit of detection (LOD) and the limit of quantification (LOQ), as well as R^2^ as a measure for the goodness of the fit, was evaluated (see Table 3). 

The FAIMS error bars shown in Figure 4a are relatively big. Several different sources have presumably contributed to the acetone signal variance. First, three slight features apart from RIP, monomer and dimer peak were visible in the dispersion plots of the acetone/CO_2_/air sample (see Figure 3c at ~65% DF / −3 V CV, at ~50% DF / −0.5 V CV, and at ~75% DF / +0.5 V CV). As mentioned before, these possibly appeared due to contaminations from the CO_2_ gas bottle and due to evaporations from the tubing and the FAIMS itself. Even if they do not seem to have fundamentally impacted the obtained data, these contaminations might still have competed with acetone for the ionization energy in the FAIMS ionization region, therefore possibly altering the acetone signal intensity and increasing the associated error bars. Furthermore, it cannot be excluded that slight humidity variations occurred, additionally enhancing the variance of the acetone signal. Finally, the saturation of the FAIMS detector at higher acetone concentrations can be assumed to also have made a contribution to the signal variance. 

### 3.2. Evaluation of the Hybrid FAIMS-FTIR-LS Setup

#### 3.2.1. Feasibility of the Integration of FAIMS, iHWG-FTIR, and LS into a Single System

Despite the significant differences in measurement protocols (i.e., in terms of background recording, duration of an individual measurement, etc.) and experimental requirements (i.e., gas flow and pressure, etc.), the integration of FAIMS, iHWG-FTIR spectroscopy and LS into a single system was successful. Synthetic breath samples supplied by the gas mixing system described in Section 2.1.1 were analyzed online without any prior sample preparation. 

With regard to the future goal of implementing a hybrid FAIMS-FTIR-LS system in a MICU for analyzing minute volumes of exhaled breath from traumatized mice, it was relevant that cumbersome sample handling, such as injecting sample aliquots into each analytical device individually, along with associated handling errors, are inherently and successfully avoided. Also, integration into a single system decreases the dead volume and increases the compactness of the hybrid device, which is essential due to the limited instrumental space in a MICU.

Furthermore, the ability to conduct online measurements will enable quasi-continuous and close to real-time monitoring of the mouse’s physiological status (e.g., RQ) with the data sampling frequency only limited by the duration of the measurement. Currently, the duration for recording a single FAIMS dispersion plot (~4 min) by far exceeds the duration of the FTIR and LS measurements (~40 and 1 s, respectively). Consequently, approximately 15 breath data points per hour may currently be recorded. However, the time required for a FAIMS measurement may be significantly reduced in the future by exclusively recording data in the relevant CV and DF range, i.e., only the range where monomer and dimer peak data are evaluated (−2.75 to +0.45 V CV and 46 to 72% DF). As a result, a FAIMS measurement duration of approximately 20 s is anticipated. By reducing the spectral resolution of the FTIR measurement, the duration of an individual spectra acquisition may be reduced to approximately 20 s (i.e., similar to FAIMS). Hence, in the future, a data sampling frequency of 180 data points per hour comprising all three orthogonal data sets is anticipated. This is especially remarkable when compared to the data rates currently available via GC-based analysis in our MICU, which provide one data point per hour recorded offline via GC-MS [48].

#### 3.2.2. Detection of the Target Analytes within Breath-Relevant Concentration Ranges 

FAIMS, FTIR and LS signal in dependence on the acetone and the CO_2_ concentration are displayed in Figure 4b,c.

It is evident that the detection of all three target analytes was successful in their breath-relevant concentration ranges [56]: While CO_2_ and O_2_ concentrations in the respective typical, low, and medium percentage ranges could be easily quantified, detection of acetone as a representative for breath VOCs in the low ppb range was also enabled, even down to the low to medium ppt range (see Table 3). 

#### 3.2.3. Orthogonality of FAIMS, iHWG-FTIR, and LS

Orthogonal analytical methods determine analytes based upon interaction with different physical stimuli/detection principles, thereby addressing different groups of analytes in a complementary fashion. However, true orthogonality also prerequisites that the analytes and their individual detection do not affect or interfere with their respective analytical signals, i.e., their signals are mutually independent. 

In the present case, FAIMS, iHWG-FTIR spectroscopy and LS each only detected one of the selected target analytes, respectively. Hence, none of the three methods could have been omitted without losing information about the molecular composition of a sample: O_2_ detection via LS was necessary, as O_2_ is neither IR active nor did it give rise to a FAIMS signal. Furthermore, CO_2_ was not detected by the luminescence sensor and is not ionizable by the ^63^Ni FAIMS ionization source, and hence, was not detected by FAIMS. However, it gave rise to an iHWG-FTIR signal. Last but not least, the luminescence sensor did not respond to acetone, and the sensitivity of the selected IR approach did not allow for acetone detection in the tested breath-relevant concentration range, even though acetone, in principle, is IR-active. Integrating FAIMS into the diagnostic platform was thus essential for reliable and sensitive trace acetone detection.

Even if each analyte only gave rise to a signal in one of the detection methods, there could still be interference of the signals, e.g., by collisions between gas molecules in iHWG-FTIR spectroscopy or by reaction with the sensing dye in LS. However, it was demonstrated that indeed the signals were mutually independent of each another (see Figure 4). The acetone signal was statistically identical, regardless if the CO_2_ concentration was 3%, 4%, or 5% (Figure 4a). Likewise, the according analytical figures of merit, i.e., LOD, LOQ, R^2^ and parameters of the asymptotic fit, did not depend on the CO_2_ concentration (see Table 3). Hence, the CO_2_ concentration did not have any effect on the FAIMS results. Reversely, Figure 4b and 4c reveal, that the acetone concentration did neither affect the CO_2_ nor the O_2_ signal. Also, the O_2_ signal did not change depending on the CO_2_ concentration, but stayed constant irrespective if 3%, 4%, or 5% CO_2_ were present. In conclusion, no mutual co-dependencies of the acetone, CO_2_ and O_2_ signal were detected. This underlines the excellent orthogonality of FAIMS, FTIR spectroscopy and LS, making their combination especially suitable for a complex matrix like exhaled breath: simply by selecting this suitable combination of analytical methods, a first—at least virtual—“preseparation” of the sample components has been undertaken, thus already simplifying the analytical task. 

### 3.3. Towards Real Breath Analysis

Even though the results of this feasibility study are promising, several challenges still have to be overcome prior to enabling online exhaled breath analysis in a MICU. Figure 5 gives an overview on the achievements of the present study and the remaining challenges towards real-world implementation. The challenges that are going to be addressed in the near future are explained in more detail thereafter.

#### 3.3.1. Pre-Separation of Analytes

Unlike in our model samples, of course more than one VOC is present in real breath. All contained breath VOCs will compete for the FAIMS ionization energy, and therefore, will cause co-dependencies of their signals. To prevent this, pre-separation based on a GC or an MCC column will be integrated into the hybrid setup enabling the VOCs to reach the ionization region one-by-one. Since the contaminations discussed above (see Figure 3c) will also be separated from the analytes via the GC or MCC column, the FAIMS signal variance may, in fact, additionally benefit in analytical quality via such pre-separation schemes. 

#### 3.3.2. Effects of Humidity 

The samples tested to date only contained minimal amounts of water, whereas real breath is oversaturated with humidity. Although FAIMS analysis is possible even at extremely humid conditions, the presence of high–and potentially varying–amounts of humidity will have a major effect on the FAIMS signal intensity, since the FAIMS detection mechanism is based on ionized water clusters [57,58]. Therefore, chemometric data treatment in dependence of the present water level or experimentally filtering out the humidity by a condenser as proposed by Maiti et al. [59], which is explicitly suitable for dehumidifying breath without significant VOC loss, could be possible strategies to address high humidity levels in breath. 

#### 3.3.3. Alkane Detection

Alkanes are an important class of breath VOCs [21], which cannot be detected with the current system, as they are not ionized by a ^63^Ni ionization source. This could be addressed by taking advantage of the FTIR detection unit. When extending the optical path length of the iHWG and replacing the FTIR spectrometer by a more intense IR light source, such as tunable quantum or interband cascade laser, the LOD/LOQ for alkane detection via FTIR could be lowered towards breath-relevant concentrations. The additional implementation of preconcentration schemes may further complement this strategy. 

## 4. Conclusions

The long-term goal of the present study is to develop hybrid analytical platforms comprising FAIMS, iHWG-FTIR spectroscopy, and LS for online monitoring of CO_2_, O_2_, and volatile organic compounds, including alkanes in exhaled mouse breath. The results presented herein demonstrate the feasibility of this concept by proving that despite different experimental requirements and measurement protocols, the integration of these methods into a single compact system, as well as online measurements, are indeed possible. Furthermore, all three target analytes—CO_2_, O_2_, and acetone serving as a representative for VOCs—could be detected in their respectively breath-relevant concentration ranges. Last but not least, FAIMS, iHWG-FTIR and LS exhibited excellent orthogonality, i.e., each of the three methods only responded to one of the target analytes and the recorded analytical signals were not affected by the presence of the other analytes. Consequently, the presented hybrid prototype integrating FAIMS, iHWG-FTIR and LS into a single system is a promising step towards hyphenated analytical tools expanding the measurement capabilities in modern exhaled breath analysis. 

However, the presented method still has to resolve some limitations prior to implementation in real-world scenarios such as mouse intensive care units. So far, the system has only been tested via synthetic breath samples containing only one VOC and trace amounts of water, which is clearly less complex vs. real breath. During future optimization studies, the system will be tested using synthetic breath samples containing more than one VOC and higher humidity levels, as well as with real breath samples. Since this will induce interferences within the signals of VOCs detectable by FAIMS, a pre-separation scheme prior to FAIMS analysis should be included. Also, the influence of high humidity levels will have to be evaluated and corresponding solutions (e.g., either removal of water prior to analysis, multivariate data treatment taking humidity as a variable into account, etc.) will have to be integrated. Also, the detection of alkanes (e.g., ethane, pentane) will have to be enabled with an improved sensitivity of the iHWG-FTIR module achievable by extending the optical path length and implementing IR laser light sources. Finally, a suitable software solution will replace the Matlab script for providing a user-friendly system at the MICU.

## Figures and Tables

**Figure 1 sensors-19-02653-f001:**

Motivation behind the development of a combined FAIMS-FTIR-LS breath analysis platform.

**Figure 2 sensors-19-02653-f002:**
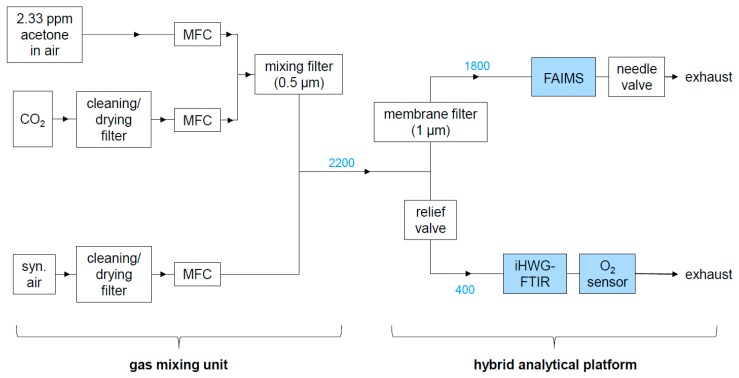
Experimental setup comprising the gas mixing unit and the hybrid analytical platform. Numbers in blue are gas flows in mL/min.

**Figure 3 sensors-19-02653-f003:**
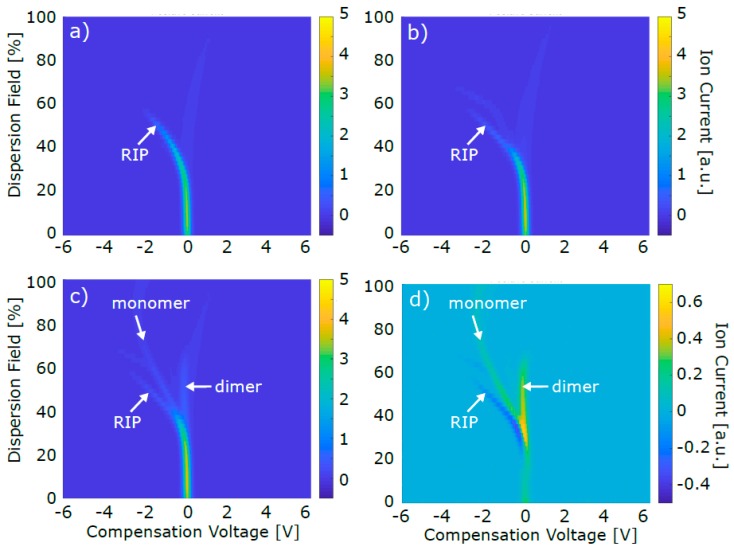
FAIMS dispersion plots. (**a**) Pure air sample (**b**) CO_2_/air sample (**c**) acetone/CO_2_/air sample (**d**) background subtracted acetone/CO_2_/air sample ((**c**) minus (**b**)). CO_2_ and acetone concentration of these exemplary data were 4% and 1 ppb, respectively. For the sake of clarity, not all four graphs wear all three axes labels.

**Figure 4 sensors-19-02653-f004:**
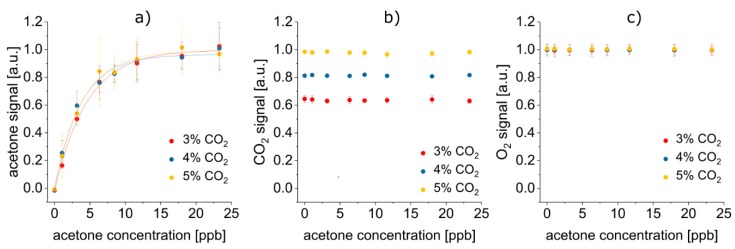
No mutual signal co-dependencies of acetone, CO_2_, and O_2_ were detected. All displayed error bars are 1σ error bars. (**a**) Acetone signals recorded with FAIMS depend on the acetone concentration (asymptotic fit *y* = *A* − *B*·*C_x_*), yet, remains independent of the CO_2_ content. (**b**) CO_2_ signals recorded by iHWG-FTIR only vary depending on the CO_2_ concentration. (**c**) O_2_ signals recorded by LS are neither influenced by the acetone nor by the CO_2_ content.

**Figure 5 sensors-19-02653-f005:**
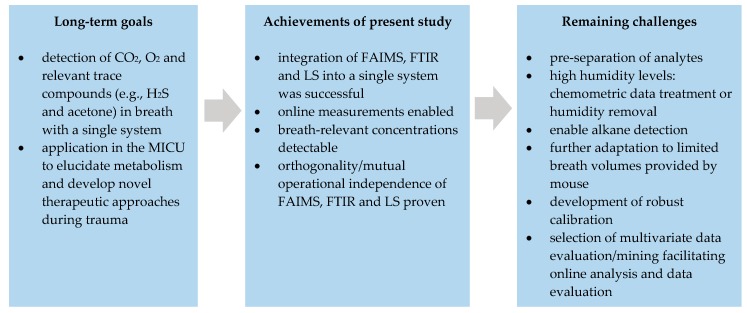
Overview on the long-term goals, the achievements of the present feasibility study and the remaining challenges prior to implementation of the hybrid system into a MICU.

**Table 1 sensors-19-02653-t001:** Overview on hyphenated techniques for exhaled breath analysis.

Breath Analysis Technique	Target Analytes	Multiple/Ortho-Gonal Detection? ^A^	Integrated Setup? ^B^	Online Monitoring? ^C^	Comment	Reference
detector coupled with preconcentration and/or preseparation, e.g., GC-MS	various VOCs and inorganic compounds	□	n.a.	■	limited scope of target analytes due to usage of only detector type; MS addresses many analytes, but is bulky and costly [7]	[3,4,14,15,16,17,18,19]
GC-IMS & electronic nose	various VOCs and inorganic compounds (e.g., NH_3_, SO_2_)	■	□	□ enose ■ GC-IMS	^D) E)^	[25]
isotope ratio MS & SIFT-MS & Quintron breath tracker	VOCs, H_2_, CH_4_, ^13^CO_2_	■	□	□	^D) E)^	[26]
SIFT-MS & Gastrocheck Instrument	VOCs, H_2_, CH_4_,	■	□	□	^D) E)^	[27]
APCI-MS & IMS	VOCs, CO	■	□	□	^D) E)^	[28]
PTR-TOF-MS & NDIR gas analyzer	VOCs, CO_2_	■	□	■	^D)^	[29]
GC-MS, GC-IMS, electronic nose & NDIR/luminescence quenching	VOCs, CO_2_, O_2_	■	■ CO_2_,O_2_ □ VOCs	■ CO_2_, O_2_ □ VOCs	^D E^ (for VOCs)	[30]
GCxGC-FID/qMS	various VOCs	■	■	□	^E)^	[31]
electronic noses, e.g., Cyranose 320	various VOCs and inorganic compounds	■/□ (depending on sensor choice)	■	■	portable and cost-efficient; biomarker identification and inter-device comparability limited	[32,33,34,35,36,37,38,39]
multi-wavelength range spectrometry	various (e.g., ethanol, H_2_O_2_ [41]; e.g., NO, CO [42])	□	■	■ [42] no info [41]	usage of multiple wavelength ranges provides some orthogonality, but basic detection principle remains the same (i.e., absorption of light)	[41,42]
PTR-TOF-MS, spirometry, capnometry & hemodynamics	VOCs, CO_2_, O_2_	■	■	■	adapted for human breath; large dead volume between analytical setup and patient would impair mechanical ventilation of small animals such as mouse	[43,44]

^A^ Provides information whether analyte detection is based on one (□) or on various (■) (orthogonal) detection principles. ^B^ Provides information whether the used detection units were used as standalone (□) techniques or whether they were integrated into a single setup (■). ^C^ Provides information whether online analysis (■) was enabled or whether all analysis was conducted offline (□). ^D^ Using standalone techniques can require laborious sample handling and extended analysis times, risking handling errors and limiting the usefulness of the setup for POC applications. ^E^ Offline analysis entails gas bag usage and/or sample storage, potentially giving rise to cross-contamination and sample degradation [3]. For the sake of readability, abbreviations mentioned in this table for the first time are not explained within the table. Please refer to the Abbreviations section at the end of the manuscript.

**Table 2 sensors-19-02653-t002:** Summary of the experimental details when operating the hybrid FAIMS-FTIR-LS system.

		FAIMS	iHWG-FTIR	LS
**Ana-lytes**	**Target analyte**	acetone	CO_2_	O_2_
	**Concentration**	0–23 ppb	3%–5%	19.6% ± 0.5%
**Sample type/** **meas. protocol**	**pure air**	verifying system cleanliness	background recording	-
	**CO_2_/air**	background recording	-	-
	**acetone/CO_2_/air**	acetone signal recording	CO_2_ signal recording	O_2_ signal recording
	repeated for all eight samples of a measurement series in random order			
**Device details and** **data evaluation**	**device**	OEM FAIMS PAD (owlstone)	ALPHA FTIR spectrometer (Bruker)	FireStingO2 (PyroScience)
	**measurement principle**	diff. ion mobility at high and low field	abs. of vibration-inducing IR radiation	quenching of organic dye luminescence
	**flow [mL/min]**	1800 ± 30	400 ± 10	
	**pressure [bar*g*]**	0.800 ± 0.020	0	
	**duration of measurement**	19 min (for 5 replicates)	3.5 min (for 5 replicates)	10 min (for 600 data points)
	**analyte signal**	peak volume sum for monomer and dimer	area under peak centered at 2360 cm^−1^	signal intensity as output by sensor

**Table 3 sensors-19-02653-t003:** Analytical figures of merit of the concentration-dependent FAIMS measurements of acetone. No statistical difference between fit parameters A, B and C of the asymptotic fit (equation *y* = *A* − *B*·*C^x^*) at 3%, 4%, or 5% CO_2_. R^2^, concentration at LOD and concentration at LOQ varied, yet with no clear trend visible depending on the CO_2_ content. This indicates independence of the acetone signal from the CO_2_ concentration.

	3% CO_2_	4% CO_2_	5% CO_2_
fit parameter A	0.998 ± 0.019	0.964 ± 0.025	0.989 ± 0.023
fit parameter B	1.024 ± 0.025	0.962 ± 0.038	0.997 ± 0.034
fit parameter C	0.803 ± 0.012	0.765 ± 0.022	0.772 ± 0.019
R^2^ >	0.995	0.989	0.992
LOD [ppt]	145	78	56
LOQ [ppt]	358	405	165

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
