# Peer review of "Hybrid Analytical Platform Based on Field-Asymmetric Ion Mobility Spectrometry, Infrared Sensing, and Luminescence-Based Oxygen Sensing for Exhaled Breath Analysis"

_sensors, 2019, doi:10.3390/s19122653_

Round 1
Reviewer 1 Report
References in lines 47-49, should present the recently published review papers.
Please consider to deliver a Fig. to the Introduction part to summarized the reviewed methods presented by other researchers. The review is not-well prepared; in the present form it is not clear how the proposed method is better/novel in comparison with the state of art. I suggest a Figure, it can be used as Graphical Abstract as well.
Reviewer 2 Report
The authors present comparison of techniques for the analysis of compounds on exhaled breath. The techniques include FAIMS and Infra red sensors and luminescence based oxygen sensing. There is a need for studies that compare techniques in a systematic way for breath analysis and to improve knowledge in this expanding area.
The abstract of the paper could be improved by adding some specific details about the exact findings, I do not get a sense of what was discovered during this work, but I get a sense of what is still required to be undertaken. A succinct summary of the results for each techniques and thei comparison is required.
Introduction
There are some good reviews of breath constituents/compounds in health and disease by Amann and co workers which could be added to the introduction to better set the challenges of breth research.
The experimental detail is well outlined in the manuscript
Results section
THe following statement " It could not be erased throughout the whole project and was likely
to be caused by substances emitted from the tubings and the FAIMS device itself." This is an important point and requires further explanation or work to ascertain what the feature was, I think it is well accepted that FAIMS is very sensitive and therefore, it is difficult to eradicate ll background "contamination", but what is important is how the stability of this background contamination was monitored and what measures were made to assess its affects on the results ie. blank subtraction, assessing standards etc.
Figure 2 legend - this statement is not clear " It could not be erased throughout the whole project and was likely to be caused by substances emitted from the tubings and the FAIMS device itself."
the following statement makes me question why this type of separation was not used for the model system? "First, unlike in our model samples, of course 316
more than one VOC is present in real breath. All these breath VOCs will compete for the FAIMS
ionization energy and therefore cause co-dependencies of their signals. To prevent this, preseparation based on a GC or an MCC column will be integrated into the hybrid setup, enabling the VOCs to reach the ionization region one by one."
Actually from my perspective the results section seems uncertain and is quite conversational, it needs to state the results and qualify the observations of the current study.
THe following statement calls into question the selection of the technique, however, there are plenty of examples of high humidity applications of FAIMS, so are the authors saying it is the change in humidity which is the biggest issue "Since the FAIMS detection mechanism is based on ionized water clusters, changes in humidity have a major effect on the FAIMS signal intensity. Here, chemometric data treatment in dependence of the present water level or experimentally filtering out the humidity by a condenser as proposed by Maiti et al.[35], which is explicitly suitable for dehumidifying breath without significant VOC loss, could be possible solutions."
I think if the authors could carefully review the manuscript to address these points and generally improve the manuscript to make the message more clear and concise then it could be published
Reviewer 3 Report
The paper has merit and deserves attention. Furthermore, it fits the scope of the Journal it was submitted to. However, several issues should be considered before its acceptance, as mentioned below.
Introduction:
The inclusion of more recent SOA would benefit the quality of the article. Please, check and include recent works (articles, reviews…), especially concerning the e-nose systems (e.g., Mazzatenta et al., 2013 Adv Exp Med Biol., Ritz et al., 2016 Biol Psychol., Tonacci et al., 2018 Int J Psychol., or Tonacci et al., 2019 Biosensors)
Materials and Methods:
This part is very well explained. However, to increase its readability, I would suggest a more schematic presentation of the procedure (e.g., by placing it in a Table or Figure)
Results:
How were you able to demonstrate the orthogonality of the results obtained? Please, briefly discuss.
In the perspective of a real online breath analysis, did you consider also changing the analysis methods in terms of software selection, since Matlab is not the most user-friendly option to this extent? Please, discuss.
Conclusions:
A more practical conclusion, including a stronger take-home message and future development should be added. In addition, limitations of the current approach should be clearly acknowledged.
Some typos are present throughout the text. Please, check.
Round 2
Reviewer 2 Report
The authors have provided extensive answers to my questions and have made considerable changes to the manuscript, overall I am satisfied that they have answered my points. Therefore, I believe the manuscript which describes an interesting topic is worthy of publication in sensors.
Reviewer 3 Report
In my opinion, the paper has been significantly improved, it is more scientifically sound and excellent in its form. I just have a very minor concern at this stage, that is the inclusion of the software selection concerns in the limitations of this work (generating interesting tips for future development).
